aerospace

MDPI

*Article*

# Shock Equations and Jump Conditions for the 2D Adjoint Euler Equations

**Carlos Lozano * and Jorge Ponsin**

Computational Aerodynamics Group, National Institute of Aerospace Technology (INTA), Carretera de Ajalvir, km 4, 28850 Torrejón de Ardoz, Spain
* Correspondence: lozanorc@inta.es

**Abstract:** This paper considers the formulation of the adjoint problem in two dimensions when there are shocks in the flow solution. For typical cost functions, the adjoint variables are continuous at shocks, wherein they have to obey an internal boundary condition, but their derivatives may be discontinuous. The derivation of the adjoint shock equations is reviewed and detailed predictions for the behavior of the gradients of the adjoint variables at shocks are obtained as jump conditions for the normal adjoint gradients in terms of the tangent gradients. Several numerical computations on a very fine mesh are used to illustrate the behavior of numerical adjoint solutions at shocks.

**Keywords:** adjoint Euler equations; shocks; normal derivatives



**Citation:** Lozano, C.; Ponsin, J. Shock Equations and Jump Conditions for the 2D Adjoint Euler Equations. *Aerospace* **2023**, *10*, 267. https://doi.org/10.3390/aerospace10030267

Academic Editor: Konstantinos Kontis

Received: 13 February 2023
Revised: 6 March 2023
Accepted: 8 March 2023
Published: 10 March 2023



## 1. Introduction

This work deals with some technical details of inviscid adjoint equations in the presence of shocks. Understanding how inviscid adjoint solutions behave at shocks is relevant for transonic and supersonic design and control applications [1], with potential applications ranging from drag or sonic boom reduction to fluid-structure and flutter control, but also for error analysis and correction [2]. To understand the adjoint approach for shocked flows, three major points need to be addressed.

1. The correct formulation of the inviscid adjoint equations at shocks must be addressed, which has to consider linearized perturbations to the shock location. This analysis was carried out in [3,4] for the quasi-1D adjoint Euler equations and in [5,6] for the 2D adjoint Euler equations. In both cases, it was shown that, for typical cost functions, the adjoint variables are continuous at shocks where they have to obey an adjoint boundary condition. The correct formulation and approximation of adjoint equations in flows with shocks has also been addressed in [7–9].
2. The accuracy of discretized adjoint approximations with shocks must also be considered. For numerical computation, the adjoint shock boundary condition is usually not explicitly enforced, but the discretized adjoint equations (either continuous or discrete) yield the correct solution as long as adequate levels of smoothing are applied, in such a way that the shock is spread over an increasing number of grid points as the mesh spacing is reduced [10–12]
3. The last point to address is the assessment of the impact of the shock equations on practical applications such as aerodynamic design or error estimation, among others. In numerical implementation, especially in 2D, such shock conditions and increased smoothing are usually ignored under the assumption that they have little impact on both the adjoint solution and the sensitivities. However, results in [6] show that explicitly accounting for the adjoint shock condition can have a significant impact on both the sensitivities and the optimization procedure. For adjoint-based error correction with shocks, it was shown in [2] with 1D examples that the key to obtaining meaningful results is the use of discrete solutions with a well-resolved viscous shock.

The focus of the present work will be on point 1. While in quasi-1D the behavior of the adjoint variables and their derivatives can be completely established from the shock and adjoint equations [3,4,13], a similar analysis for the 2D case is not yet fully available. It was argued in [5] that the adjoint variables, relative to an output function consisting of the integral of a function of the pressure along the airfoil surface, are continuous across shocks and that an adjoint boundary condition is required along the length of the shock. It was also claimed that gradients of adjoint variables are discontinuous, but no proof or conclusive numerical result was offered. A complete analysis of the 2D case involved was presented in [6], which showed that, for typical costs functions (aerodynamic lift and drag, for example), the adjoint variables are continuous across shocks and that the adjoint variables obey a differential equation along the shock. The former statement is cost-function dependent. For example, entropy variables are adjoint states relative to a cost function measuring the net entropy flux across the domain boundaries (including the shock surface) [14] and are discontinuous across shocks.

Concerning the behavior of the gradients of adjoint variables at shocks, no further analysis was conducted in [6]. A first step in that direction was taken in [13], which presented the jump conditions for the gradients of the adjoint variables at shocks. It was not possible to derive closed-form results for all the derivatives, not even for normal shocks, but it was claimed, based on preliminary analysis and numerical computations, that the results were compatible with the derivatives of the adjoint variables along the shock normal direction being continuous and mostly vanishing, at least for normal shocks. However, normal shocks are difficult to realize in practice and none of the shocks presented in [13] were normal, so the conclusions offered were somewhat misleading. In [15], a series of detailed numerical experiments on very fine meshes confirmed that adjoint derivatives are indeed generally discontinuous across generic (oblique) shocks.

Here, we would like to complement the results of [13,15] in two directions. First, after reviewing the derivation of the adjoint shock equations in the 2D case, we will expand the analysis of the jump conditions of the adjoint gradients to put it on firmer grounds. Second, we will examine several numerical solutions containing oblique and normal shocks to illustrate the differences between both cases in the light of the derived shock conditions.

## 2. Adjoint Equations for Shocked Flows

This section contains a review of some known technical details of 2D adjoint equations in the presence of shocks originally derived in [6] (see also [13]). We consider steady transonic inviscid flow past a body of surface $S$ such that the flow contains a shock attached to $S$ with profile $\Sigma$ extending from $x_b$ (shock foot) to $x_{end}$ (shock tip).

We define the following cost function:

$$J(S) = \int_{S \setminus x_b} h(p)(\vec{n}_S \cdot \vec{d}) dS, \tag{1}$$

where $p$ is the pressure and $\vec{n}_S$ is the unit normal vector to $S$ (see Figure 1). When $h(p) = p$, Equation (1) corresponds to the force exerted by the fluid on $S$ measured along a direction $\vec{d}$. The flow is governed by nonlinear (steady) flow equations:

$$\nabla \cdot \vec{F}(U) = 0$$
$$U = (\rho, \rho u, \rho v, \rho E)^T \tag{2}$$

and shock equations (the Rankine–Hugoniot conditions), which for a steady shock can be written as:

$$[\vec{F} \cdot \vec{n}_\Sigma]_\Sigma = 0 \Leftrightarrow \begin{cases} [\rho v_n]_\Sigma &= 0 \\ [\rho v_n^2 + p]_\Sigma &= 0 \\ [v_t]_\Sigma &= 0 \\ [H]_\Sigma &= 0 \end{cases}. \tag{3}$$

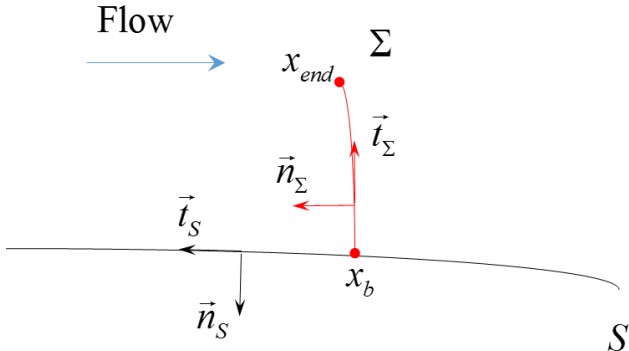

**Figure 1.** Scheme of shock location and conventions.

In Equations (2) and (3), $\vec{F} = (\rho\vec{v}, \rho\vec{v}u + p\hat{x}, \rho\vec{v}v + p\hat{y}, \rho\vec{v}H)^T$ is the flux vector, $\rho, \vec{v} = u\hat{x} + v\hat{y}, p, E, H$ are the fluid's density, velocity, pressure, total energy and enthalpy, respectively, $[\ ]_\Sigma = ()|_{downstream} - ()|_{upstream}$ denotes the jump across the shock and $v_n = \vec{v} \cdot \vec{n}_\Sigma$ and $v_t = \vec{v} \cdot \vec{t}_\Sigma$ are the velocity components in the local shock frame [16] (see Figure 1). For normal shocks, $v_t = 0$ on either side and thus the 3rd equation in Equation (3) is trivially verified.

Let us now consider the problem of computing the sensitivity derivatives of $J(S)$ given by Equation (1) with respect to, say, changes in the shape of $S$. A deformation of $S$ along the local normal direction $\vec{x}_S \to \vec{x}_S + \delta S\, \vec{n}_S$ gives rise to a fluid perturbation $\delta U$ that affects both the cost function and the shock. In the perturbed flow, the new shock shape can be described in terms of a local normal deformation $\vec{x}_\Sigma \to \vec{x}_\Sigma + \delta\,\Sigma\, \vec{n}_\Sigma$ (Figure 2).

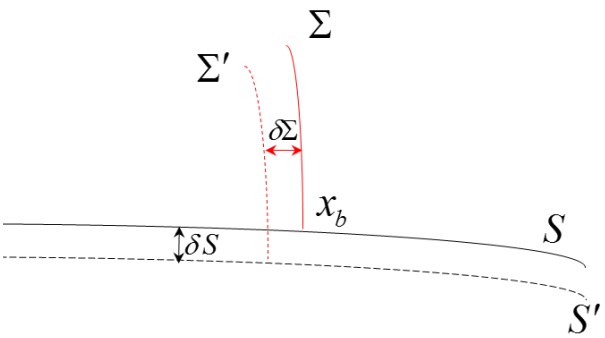

**Figure 2.** Scheme of shock perturbation.

As a result of the perturbation, the cost function changes as well, and the corresponding linearized perturbation is constrained by the following linearized flow equation:

$$\nabla \cdot (\vec{F}_U \delta U) = 0, \tag{4}$$

where $\vec{F}_U = \partial\vec{F}/\partial U$ is the flux Jacobian, and the linearized Rankine–Hugoniot conditions:

$$[\vec{F}_U \delta U]_\Sigma \cdot \vec{n}_\Sigma + [\vec{F} \cdot \delta\vec{n}_\Sigma]_\Sigma + [\delta\,\Sigma\partial_{n_\Sigma}\vec{F} \cdot \vec{n}_\Sigma]_\Sigma = 0. \tag{5}$$

The adjoint approach for this problem uses two adjoint fields: a bulk adjoint $\psi = (\psi_1, \psi_2, \psi_3, \psi_4)^T$ (the Lagrange multiplier for the Euler Equation (2)) defined in $\Omega\backslash\Sigma$, where $\Omega$ denotes the fluid domain, and the shock adjoint $\psi_s$, the Lagrange multiplier for the RH

Conditions (3), which is defined along $\Sigma$. With the aid of the adjoint fields, the cost function is reformulated as follows:

$$J(S) = \int_{S\backslash x_b} h(p)(\vec{n}_S \cdot \vec{d})ds - \int_{\Omega\backslash\Sigma} \psi^T \nabla \cdot \vec{F} d\Omega - \int_\Sigma \psi_s^T [\vec{F} \cdot \vec{n}_\Sigma]_\Sigma d\Sigma. \tag{6}$$

Linearizing Equation (6) using Equations (4) and (5) yields [6]:

$$\delta J(S) = \int_{S\backslash x_b} \delta S \vec{d} \cdot \nabla h(p)ds + \int_{S\backslash x_b} (\vec{n}_S \cdot \vec{d})\partial_p h(p)\delta p ds + \left(\frac{(\vec{n}_S \cdot \vec{n}_\Sigma)\delta S - \delta\Sigma}{(\vec{n}_S \cdot \vec{t}_\Sigma)}\right)_{x_b} [h(p)]_{x_b}(\vec{n}_S \cdot \vec{d})$$
$$-\int_{\Omega\backslash\Sigma} \psi^T \nabla \cdot (\vec{F}_U \delta U)d\Omega - \int_\Sigma \psi_s^T [\vec{F}_U \delta U]_\Sigma \cdot \vec{n}_\Sigma d\Sigma - \int_\Sigma \psi_s^T [\vec{F} \cdot \delta\vec{n}_\Sigma]_\Sigma d\Sigma - \int_\Sigma \psi_s^T [\delta\Sigma \partial_{n_\Sigma} \vec{F} \cdot \vec{n}_\Sigma]_\Sigma d\Sigma \tag{7}$$

where $\partial_{n_\Sigma} = \vec{n}_\Sigma \cdot \nabla$ is the normal derivative along the shock [16], $\delta\vec{n}_\Sigma = -\partial_{tg}(\delta\Sigma)\vec{t}_\Sigma$, where $\partial_{tg} = \vec{t}_\Sigma \cdot \nabla$ is the tangent derivative along the shock, and $[\ ]_{x_b}$ is the jump across the shock at the shock foot. In Equation (7), the third term on the first line includes the effect of a linearized displacement $\delta x_b = ((\vec{n}_S \cdot \vec{n}_\Sigma)\delta S - \delta\Sigma)/(\vec{n}_S \cdot \vec{t}_\Sigma)$ in the shock foot location (see Figure 3 and the appendix of [6] for a detailed derivation of this term).

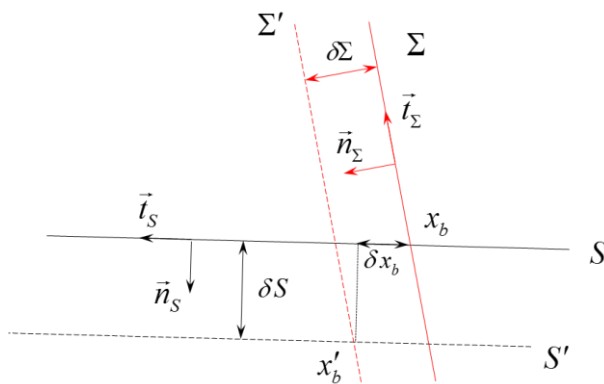

**Figure 3.** Details of shock perturbation and computation of the shock displacement term.

Integrating by parts the domain term in Equation (7) yields:

$$\int_{\Omega\backslash\Sigma} \psi^T \nabla \cdot (\vec{F}_U \delta U)d\Omega = -\int_{\Omega\backslash\Sigma} \nabla\psi^T \cdot \vec{F}_U \delta U d\Omega + \int_{S\backslash x_b} \psi^T (\vec{F}_U \cdot \vec{n}_S)\delta U ds + \int_{S^\infty} \psi^T (\vec{F}_U \cdot \vec{n}_{S^\infty})\delta U ds^\infty$$
$$+\int_\Sigma [\psi^T (\vec{F}_U \cdot \vec{n}_\Sigma)\delta U]_\Sigma d\Sigma = -\int_{\Omega\backslash\Sigma} \nabla\psi^T \cdot \vec{F}_U \delta U d\Omega + \int_{S\backslash x_b} (\vec{\varphi} \cdot \vec{n}_S)\delta p ds \tag{8}$$
$$+\int_{S\backslash x_b} \rho\delta\vec{v} \cdot \vec{n}_S(\psi_1 + \vec{\varphi} \cdot \vec{v} + \psi_4 H)ds + \int_{S^\infty} \psi^T (\vec{F}_U \cdot \vec{n}_{S^\infty})\delta U ds^\infty + \int_\Sigma [\psi^T (\vec{F}_U \cdot \vec{n}_\Sigma)\delta U]_\Sigma d\Sigma$$

where the identity $\psi^T (\vec{F}_U \cdot \vec{n}_S)\delta U = (\vec{\varphi} \cdot \vec{n}_S)\delta p + \rho\delta\vec{v} \cdot \vec{n}_S(\psi_1 + \vec{\varphi} \cdot \vec{v} + \psi_4 H)$, with $\vec{\varphi} = (\psi_2, \psi_3)$, has been used to rewrite the wall integral, and $S^\infty$ denotes the far-field boundary. Inserting Equation (8) into Equation (7) and rearranging yields:

$$\begin{aligned}\delta J(S) = \ & \int_{S\backslash x_b} \delta S \vec{d} \cdot \nabla h(p)ds + \left(\frac{(\vec{n}_S \cdot \vec{n}_\Sigma)\delta S - \delta\Sigma}{(\vec{n}_S \cdot \vec{t}_\Sigma)}\right)_{x_b} [h(p)]_{x_b}(\vec{n}_S \cdot \vec{d}) \\ & +\int_{S\backslash x_b} \rho(\delta S \partial_{n_S} \vec{v} \cdot \vec{n}_S + \vec{v} \cdot \delta\vec{n}_S)(\psi_1 + \vec{\varphi} \cdot \vec{v} + \psi_4 H)ds \\ & +\int_\Sigma \psi_s^T [\vec{F} \cdot \vec{t}_\Sigma]_\Sigma \partial_{tg}\delta\Sigma d\Sigma - \int_\Sigma \psi_s^T [\delta\Sigma \partial_{n_\Sigma} \vec{F} \cdot \vec{n}_\Sigma]_\Sigma d\Sigma \\ & +\int_{\Omega\backslash\Sigma} \nabla\psi^T \cdot \vec{F}_U \delta U d\Omega + \int_{S\backslash x_b} \left((\vec{n}_S \cdot \vec{d})\partial_p h(p) - (\vec{\varphi} \cdot \vec{n}_S)\right)\delta p ds \\ & -\int_{S^\infty} \psi^T (\vec{F}_U \cdot \vec{n}_{S^\infty})\delta U ds^\infty - \int_\Sigma [(\psi^T + \psi_s^T)(\vec{F}_U \cdot \vec{n}_\Sigma)\delta U]_\Sigma d\Sigma\end{aligned}\quad, \tag{9}$$

where the linearized wall boundary condition $\delta(\vec{v} \cdot \vec{n}_S) = \delta\vec{v} \cdot \vec{n}_S + \delta S \partial_{n_S} \vec{v} \cdot \vec{n}_S + \vec{v} \cdot \delta\vec{n}_S = 0$ has been used to rewrite the third term in Equation (9). The final step is to integrate by parts $\int_\Sigma \psi_s^T [\vec{F} \cdot \vec{t}_\Sigma]_\Sigma \partial_{tg} \delta\Sigma d\Sigma$ along $\Sigma$:

$$
\begin{aligned}
\int_\Sigma \psi_s^T [\vec{F} \cdot \vec{t}_\Sigma]_\Sigma \partial_{tg} \delta\Sigma d\Sigma = \ & \psi_s^T [\vec{F} \cdot \vec{t}_\Sigma]_\Sigma \delta\Sigma \Big|_{x_{end}} - \psi_s^T [\vec{F} \cdot \vec{t}_\Sigma]_\Sigma \delta\Sigma \Big|_{x_b} - \int_\Sigma \partial_{tg} \psi_s^T [\vec{F} \cdot \vec{t}_\Sigma]_\Sigma \delta\Sigma d\Sigma \\
& - \int_\Sigma \psi_s^T [\partial_{tg} \vec{F} \cdot \vec{t}_\Sigma]_\Sigma \delta\Sigma d\Sigma - \int_\Sigma \psi_s^T [\vec{F} \cdot \partial_{tg} \vec{t}_\Sigma]_\Sigma \delta\Sigma d\Sigma
\end{aligned}
\tag{10}
$$

Now, $[\vec{F} \cdot \vec{t}_\Sigma]_{x_{end}} = 0$, $[\vec{F} \cdot \partial_{tg} \vec{t}_\Sigma]_\Sigma = \kappa_\Sigma [\vec{F} \cdot \vec{n}_\Sigma]_\Sigma = 0$ ($\kappa_\Sigma$ is the local curvature of the shock profile) and $\partial_{tg} \vec{F} \cdot \vec{t}_\Sigma = \nabla \cdot \vec{F} - \partial_{n_\Sigma} \vec{F} \cdot \vec{n}_\Sigma$ along $\Sigma$. Hence, we can write Equation (10) as follows:

$$
\begin{aligned}
\int_\Sigma \psi_s^T [\vec{F} \cdot \vec{t}_\Sigma]_\Sigma \partial_{tg} \delta\Sigma d\Sigma = \ & -\psi_s^T [\vec{F} \cdot \vec{t}_\Sigma]_\Sigma \delta\Sigma \Big|_{x_b} - \int_\Sigma \partial_{tg} \psi_s^T [\vec{F} \cdot \vec{t}_\Sigma]_\Sigma \delta\Sigma d\Sigma \\
& - \int_\Sigma \psi_s^T [\nabla \cdot \vec{F}]_\Sigma \delta\Sigma d\Sigma + \int_\Sigma \psi_s^T [\partial_{n_\Sigma} \vec{F} \cdot \vec{n}_\Sigma]_\Sigma \delta\Sigma d\Sigma = \\
& -\psi_s^T [\vec{F} \cdot \vec{t}_\Sigma]_\Sigma \delta\Sigma \Big|_{x_b} - \int_\Sigma \partial_{tg} \psi_s^T [\vec{F} \cdot \vec{t}_\Sigma]_\Sigma \delta\Sigma d\Sigma + \int_\Sigma \psi_s^T [\partial_{n_\Sigma} \vec{F} \cdot \vec{n}_\Sigma]_\Sigma \delta\Sigma d\Sigma
\end{aligned}
\tag{11}
$$

where we have used that $[\nabla \cdot \vec{F}]_\Sigma = 0$ since $\nabla \cdot \vec{F} = 0$ on both sides of the shock. Gathering Equations (9) and (11) and rearranging yields:

$$
\begin{aligned}
\delta J(S) = \ & \int_{S \backslash x_b} \delta S \vec{d} \cdot \nabla h(p) ds + \left( \frac{(\vec{n}_S \cdot \vec{n}_\Sigma) \delta S}{(\vec{n}_S \cdot \vec{t}_\Sigma)} \right)_{x_b} [h(p)]_{x_b} (\vec{n}_S \cdot \vec{d}) \\
& + \int_{S \backslash x_b} \rho (\delta S \partial_{n_S} \vec{v} \cdot \vec{n}_S + \vec{v} \cdot \delta \vec{n}_S)(\psi_1 + \vec{\varphi} \cdot \vec{v} + \psi_4 H) ds \\
& - \left( \frac{\delta\Sigma}{(\vec{n}_S \cdot \vec{t}_\Sigma)} \right)_{x_b} [h(p)]_{x_b} (\vec{n}_S \cdot \vec{d}) - \psi_s^T [\vec{F} \cdot \vec{t}_\Sigma]_\Sigma \delta\Sigma \Big|_{x_b} - \int_\Sigma \partial_{tg} \psi_s^T [\vec{F} \cdot \vec{t}_\Sigma]_\Sigma \delta\Sigma d\Sigma . \\
& + \int_{\Omega \backslash \Sigma} \nabla \psi^T \cdot \vec{F}_U \delta U d\Omega + \int_{S \backslash x_b} \left( (\vec{n}_S \cdot \vec{d}) \partial_p h(p) - (\vec{\varphi} \cdot \vec{n}_S) \right) \delta p ds \\
& - \int_{S^\infty} \psi^T (\vec{F}_U \cdot \vec{n}_{S^\infty}) \delta U ds^\infty - \int_\Sigma [(\psi^T + \psi_s^T)(\vec{F}_U \cdot \vec{n}_\Sigma) \delta U]_\Sigma d\Sigma
\end{aligned}
\tag{12}
$$

The whole point of the adjoint approach is to define the adjoint variables in such a way that $\delta J(S)$ can be computed independently of $\delta U$ and $\delta\Sigma$. This can be achieved if the bulk adjoint state obeys the adjoint equation:

$$
\vec{F}_U^T \cdot \nabla \psi = 0
\tag{13}
$$

in $\Omega \backslash \Sigma$, with the following wall and far-field boundary conditions:

$$
\begin{aligned}
\vec{\varphi} \cdot \vec{n}_S &= \partial_p h(p)(\vec{n}_S \cdot \vec{d}) \quad \text{on } S \backslash x_b \\
\psi^T (\vec{F}_U \cdot \vec{n}_{S^\infty}) \delta U &= 0 \quad \text{on } S^\infty
\end{aligned}
\tag{14}
$$

Along the shock, the adjoint variables obey the following equations:

$$
\begin{aligned}
[(\psi^T + \psi_s^T)(\vec{F}_U \cdot \vec{n}_\Sigma) \delta U]_\Sigma &= 0 \quad \text{on } \Sigma \\
\partial_{tg} \psi_s^T [\vec{F} \cdot \vec{t}_\Sigma]_\Sigma &= 0 \quad \text{on } \Sigma , \\
\frac{[h(p)]_{x_b} (\vec{n}_S \cdot \vec{d})}{(\vec{n}_S \cdot \vec{t}_\Sigma)} + \psi_s^T [\vec{F} \cdot \vec{t}_\Sigma]_\Sigma &= 0 \quad \text{at } x_b
\end{aligned}
\tag{15}
$$

In Equation (15), $\det(\vec{F}_U \cdot \vec{n}_\Sigma)$ is proportional to $1 - M_n{}^2$, where $M_n$ is the normal Mach number. Shocks only exist for upstream normal Mach numbers strictly greater than one, $M_n > 1$. (Correspondingly, $M_n < 1$ downstream of the shock.) Therefore,

$\det(\vec{F}_U \cdot \vec{n}_\Sigma)$ is generically non-zero on either side of a shock, so the only way the first equation in (15) can hold for arbitrary values of $\delta U$ that are also independent on either side of the shock is if:

$$\psi|_{\Sigma^{up}} = -\psi_s = \psi|_{\Sigma^{down}}. \tag{16}$$

Hence, $\psi$ is continuous across the shock and, in consequence, so is the tangent derivative $\partial_{tg}\psi$. The second equation in (15) then becomes an ordinary differential equation for the bulk adjoint $\psi$ along the shock:

$$\partial_{tg}\psi^T[\vec{F} \cdot \vec{t}_\Sigma]_\Sigma = [\rho]_\Sigma v_t (\partial_{tg}\psi_1 + H\partial_{tg}\psi_4) + ([p]_\Sigma + [\rho]_\Sigma v_t{}^2)\vec{t}_\Sigma \cdot \partial_{tg}\vec{\varphi} = 0, \tag{17}$$

where $\vec{t}_\Sigma \cdot \partial_{tg}\vec{\varphi} = t_\Sigma^x\,\partial_{tg}\psi_2 + t_\Sigma^y\,\partial_{tg}\psi_3$. Finally, the last equation in (15) gives an initial condition for Equation (17) at the shock foot $x_b$:

$$\psi^T(x_b)[\vec{F} \cdot \vec{t}_\Sigma]_{x_b} = [\rho]_\Sigma v_t(\psi_1 + H\psi_4) + ([p]_\Sigma + [\rho]_\Sigma v_t{}^2)\vec{t}_\Sigma \cdot \vec{\varphi} = \frac{[h(p)]_{x_b}(\vec{n}_S \cdot \vec{d})}{(\vec{n}_S \cdot \vec{t}_\Sigma)_{x_b}}. \tag{18}$$

*Detached shocks*

For sufficiently high incoming Mach number, the flow past blunt bodies contains detached (bow) shocks ahead of the body. The shock is normal directly in front of the body and extends around it as a curved oblique shock. At a sufficient distance from the body, the shock reduces to a Mach wave at points $x_a$ and $x_b$. In this case, the adjoint problem is essentially unchanged. Along the bow shock, the adjoint state is continuous and obeys the differential Equation (17), but the matching Condition (18) is now missing, being replaced by a boundary condition $\delta\Sigma\psi^T[\vec{F} \cdot \vec{t}_\Sigma]_\Sigma\Big|_{x_a}^{x_b} = 0$ that is trivially satisfied.

*Extension to 3D*

The extension to three dimensions poses no significant complications. In the shocked case, the shock is now a surface which, for the sake of the analysis, will be taken to be a single sheet attached to the wing surface along a curve $\sigma_S$ and with an additional boundary curve $\sigma_o$ along which the shock merges with the remaining smooth flow.

It can be shown that the adjoint state is still continuous across the shock, where it obeys the following shock equations:

$$\nabla_{tg}^\Sigma \psi^T \cdot [\vec{F}]_\Sigma = 0 \tag{19}$$

on the shock surface, and

$$\psi^T[\vec{F} \cdot \hat{n}_{\sigma_S}]_{\sigma_S} = \frac{[h(p)]_{\sigma_S}(\vec{n}_S \cdot \vec{d})}{\left|\vec{n}_\Sigma \times \vec{n}_S\right|_{\sigma_S}} \tag{20}$$

along the shock foot $\sigma_S$. Here, $[\,]_{\sigma_S}$ is the jump across the shock at the shock foot, $\nabla_{tg}^\Sigma$ is the tangent gradient (the covariant derivative on the shock surface), and $\hat{n}_{\sigma_S}$ is the normal vector to the shock boundary curve $\sigma_S$ but otherwise tangent to the shock surface.

## 3. Jump Conditions for the Adjoint Gradient across a Shock

The values of the adjoint variables and their derivatives at a shock are constrained by the shock Equations (17) and (18), as well as the adjoint equations on either side of the shock, which we write in shock-oriented coordinates [16] as:

$$(A_t^\pm)^T\partial_{tg}\psi + (A_n^\pm)^T\partial_n\psi^\pm = 0, \tag{21}$$

where $A_t^\pm = \vec{F}_U^\pm \cdot \vec{t}_\Sigma$ and $A_n^\pm = \vec{F}_U^\pm \cdot \vec{n}_\Sigma$ and the superscripts $\pm$ are used to distinguish upstream and downstream quantities. Taking differences across the shock yields:

$$[A_t{}^T \partial_{tg}\psi + A_n{}^T \partial_n\psi]_\Sigma = 0. \tag{22}$$

These equations, along with (17), were analyzed and tested on a numerical solution in [13], but the meshes employed were relatively coarse. A much more thorough numerical assessment on very fine meshes was carried out later on in [15]. The following equations:

$$\begin{gathered}
[v_n\partial_n\psi_1]_\Sigma - H[v_n\partial_n\psi_4]_\Sigma = 0 \\
(\gamma-1)[n_\Sigma^x\partial_n\psi_2 + n_\Sigma^y\partial_n\psi_3]_\Sigma + \gamma[v_n\partial_n\psi_4]_\Sigma = 0 \\
[v_n(-n_\Sigma^y\partial_n\psi_2 + n_\Sigma^x\partial_n\psi_3 + v_t\partial_n\psi_4)]_\Sigma + [v_n]_\Sigma[n_\Sigma^x\partial_{tg}\psi_2 + n_\Sigma^y\partial_{tg}\psi_3]_\Sigma = 0 \\
[\partial_n\psi_1]_\Sigma + [(u+v_nn_\Sigma^x)\partial_n\psi_2 + (v+v_nn_\Sigma^y)\partial_n\psi_3 + (H+v_n^2)\partial_n\psi_4]_\Sigma + [v_n]_\Sigma v_t\partial_{tg}\psi_4 = 0
\end{gathered} \tag{23}$$

that follow from Equation (22) by elementary manipulations, were shown to be obeyed to a high degree of accuracy. Here, we would like to take a step forward and try to constrain as much as possible the values of the jumps of the adjoint derivatives across the shock. The resulting equations are exact, and numerical testing should be seen more as a test on the extent to which numerical solutions obey them than as a test of the equations themselves.

It was argued above that the determinant $\det(A_n^\pm) \neq 0$ at generic points of 2D shocks, so Equation (21) can be used to solve for the normal adjoint derivatives in terms of the tangent adjoint derivatives:

$$\partial_n\psi^\pm = -(A_n^\pm)^{-T}(A_t^\pm)^T\partial_{tg}\psi^\pm. \tag{24}$$

Taking differences across the shock in Equation (24) gives equations for the jumps $[\partial_n\psi]_\Sigma$ of the normal derivatives that, using the RH conditions and the adjoint shock Equation (17), can be written as follows:

$$\begin{aligned}
[\partial_n\psi]_\Sigma &= -[(A_n)^{-T}(A_t)^T]_\Sigma\partial_{tg}\psi = \\
&- \frac{[v_n^{-1}]_\Sigma}{(\gamma-1)H+v_t^2}
\begin{pmatrix}
v_t(v_t{}^2\partial_{tg}\psi_1 - (\gamma-1)H^2\partial_{tg}\psi_4) \\
(H(\gamma-1)-v_t{}^2)(\partial_{tg}\psi_1 + H\partial_{tg}\psi_4)t_\Sigma^x \\
(H(\gamma-1)-v_t{}^2)(\partial_{tg}\psi_1 + H\partial_{tg}\psi_4)t_\Sigma^y \\
v_t(v_t{}^2\partial_{tg}\psi_4 - (\gamma-1)\partial_{tg}\psi_1)
\end{pmatrix}.
\end{aligned} \tag{25}$$

Similarly, multiplying Equation (24) by $v_n$ and taking differences across the shock yields:

$$[v_n\partial_n\psi_1]_\Sigma = 0 = [v_n\partial_n\psi_4]_\Sigma. \tag{26}$$

The manipulations required to tackle Equations (24)–(26) are carried out with the aid of a symbolic manipulation software. From Equation (25), it is possible to obtain two additional properties for the jumps:

$$\begin{gathered}
[n_\Sigma^x\partial_n\psi_2 + n_\Sigma^y\partial_n\psi_3]_\Sigma = 0 \\
[\partial_n\psi_1]_\Sigma + H[\partial_n\psi_4]_\Sigma + v_t[t_\Sigma^x\partial_n\psi_2 + t_\Sigma^y\partial_n\psi_3]_\Sigma = 0
\end{gathered}. \tag{27}$$

Notice that, according to Equation (25), the jumps in the normal adjoint derivatives are generally not zero, but they are zero if the adjoint solution is constant along the shock. This is in agreement with the fact that, for example, there is no discontinuity of the adjoint variables or their gradients across the fish-tail shocks in supersonic cases, since the adjoint variables are constant (actually zero) on either side of the shock (see Section 4.1).

Finally, for cost functions that depend only on pressure (lift and drag, for example), we have $\psi_1 = H\psi_4$ [5] and, thus, using Equation (17) we get:

$$H^{-1}\partial_{tg}\psi_1 = \partial_{tg}\psi_4 = -\frac{([p]_\Sigma + [\rho]_\Sigma v_t{}^2)}{2[\rho]_\Sigma v_t H}\left(\vec{t}_\Sigma \cdot \partial_{tg}\vec{\varphi}\right) \tag{28}$$

(where it has been assumed here that the flow is homenthalpic) and:

$$[\partial_n \psi]_\Sigma = \frac{(v_t{}^2 - H(\gamma-1))[v_n^{-1}]_\Sigma}{\gamma+1} \begin{pmatrix} 1 \\ -\frac{2}{v_t}t_\Sigma^x \\ -\frac{2}{v_t}t_\Sigma^y \\ \frac{1}{H} \end{pmatrix} \left( \vec{t}_\Sigma \cdot \partial_{tg}\vec{\varphi} \right). \tag{29}$$

It is clear from Equation (29) that the adjoint derivatives are discontinuous iff $\vec{t}_\Sigma \cdot \partial_{tg}\vec{\varphi} = 0$.

*Normal shocks*

When $v_t = 0$, we have a normal shock. Examples of normal shocks in 2D include the tip of a bow shock, as well as the detached shock in front of a sufficiently wide wedge. Normal shocks are also present in most supersonic inlets. On the other hand, the shock forming over an airfoil in transonic flow is generally not normal. If flow separation after the shock is not considered, such shocks are then normal at their feet, but staying normal to some distance above the surface (a so-called normal shock with normal extension [17]) is only possible for one very specific upstream wall Mach number, which is equal to $M^* \approx 1.662$ for $\gamma = 1.4$.

For a normal shock, the shock Equation (17) reduces to:

$$\partial_{tg}\psi^T [\vec{F} \cdot \vec{t}_\Sigma]_\Sigma = [p]_\Sigma (t_\Sigma^x \, \partial_{tg}\psi_2 + t_\Sigma^y \, \partial_{tg}\psi_3) = 0 \tag{30}$$

and, thus:

$$t_\Sigma^x \, \partial_{tg}\psi_2 + t_\Sigma^y \, \partial_{tg}\psi_3 = 0, \tag{31}$$

while the Condition (18) yields (recall that in this case $\vec{n}_S = -\vec{t}_\Sigma$ according to our conventions in Figure 1):

$$\psi^T(x_b)[\vec{F} \cdot \vec{t}_\Sigma]_{x_b} = [p]_\Sigma (\vec{t}_\Sigma \cdot \vec{\varphi}) = -[p]_\Sigma (\vec{n}_S \cdot \vec{\varphi}) = -[h(p)]_{x_b}(\vec{n}_S \cdot \vec{d}), \tag{32}$$

which, for $h(p) = p$, is simply the adjoint wall b.c. $\vec{n}_S \cdot \vec{\varphi} = \vec{n}_S \cdot \vec{d}$.

As for the remaining conditions on the normal derivatives, we get from Equation (25) (setting $v_t = 0$) the following jump conditions:

$$\begin{aligned} [\, \partial_n \psi_1]_\Sigma &= 0 \\ [\partial_n \psi_2]_\Sigma &= -\, [v_n{}^{-1}]_\Sigma (\partial_{tg}\psi_1 + H \, \partial_{tg}\psi_4) t_\Sigma^x \\ [\partial_n \psi_3]_\Sigma &= -\, [v_n{}^{-1}]_\Sigma (\, \partial_{tg}\psi_1 + H\partial_{tg}\psi_4) t_\Sigma^y \\ [\partial_n \psi_4]_\Sigma &= 0 \end{aligned} \quad . \tag{33}$$

We can actually derive stronger conditions for the normal derivatives themselves by going back to the adjoint Equation (21), setting $v_t = 0$, and solving for the normal adjoint derivatives in terms of the tangent adjoint derivatives also using Equation (31), which yields:

$$\partial_n \psi^\pm = \begin{pmatrix} 0 \\ -\partial_{tg}\psi_3 - t_\Sigma^x (\partial_{tg}\psi_1 + H \, \partial_{tg}\psi_4)/v_n^\pm \\ \partial_{tg}\psi_2 - t_\Sigma^y (\partial_{tg}\psi_1 + H \, \partial_{tg}\psi_4)/v_n^\pm \\ 0 \end{pmatrix}, \tag{34}$$

from which we get:

$$\begin{aligned} \partial_n \psi_1^\pm &= 0 \\ n_\Sigma^x \partial_n \psi_2^\pm + n_\Sigma^y \partial_n \psi_3^\pm &= 0 \\ \partial_n \psi_4^\pm &= 0 \end{aligned} \quad . \tag{35}$$

## 4. Numerical Tests

We will now present several test cases in order to illustrate the behavior of typical numerical adjoint solutions at shocks. We do not attempt to investigate the conditions under which the numerical solutions converge to the analytic solution, nor do we attempt to compare different numerical schemes as regards the behavior of the computed adjoint solutions, but rather we would like to show how a typical, finite-volume adjoint solver behaves at shocks without enforcing any shock condition.

We will be using the unstructured, open source SU2 code [18] for numerical testing. The computations are carried out on a very fine unstructured mesh obtained from the basic triangular Euler mesh (Figure 4), available at the SU2 Tutorial Collection [19], with five rounds of uniform refinement. At each round, every edge is bisected and the resulting nodes are joined to form new triangles. In order to preserve the surface of the airfoil, a Bézier-spline surface reconstruction on the basis of the previous mesh has been performed at each stage. The final mesh has 6400 nodes on the airfoil profile and $5.2 \times 10^6$ nodes and $10.4 \times 10^6$ triangular elements throughout the flowfield. The near-wall distance is around $10^{-5}$ chord lengths, which should be more than adequate to resolve the Euler flow and adjoint fields. Notice that this compares well to the current state of the art on these types of mesh-converged Euler computations [15,20]. The flow and drag-based adjoint solutions are computed with the SU2 direct and continuous adjoint solvers using a central scheme with JST artificial dissipation.

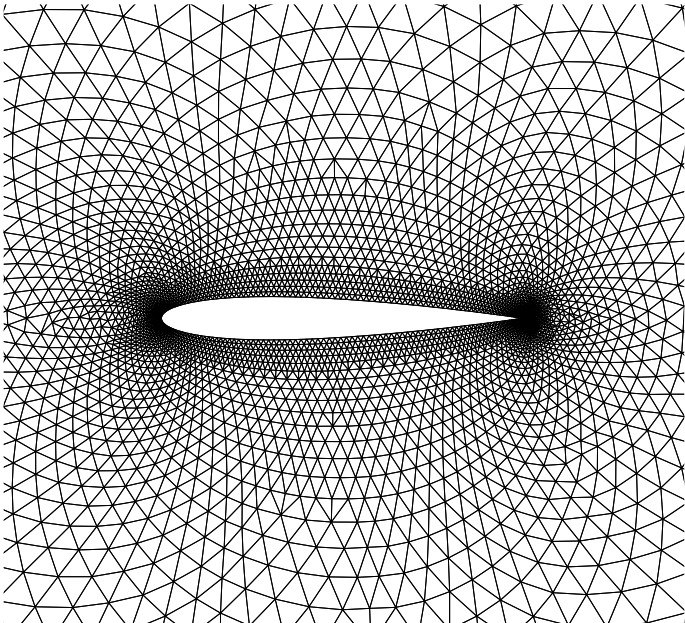

**Figure 4.** Close-up of the baseline NACA0012 mesh.

### 4.1. Supersonic Flow

Our first test case is supersonic flow past a NACA0012 airfoil. Flow conditions are $M = 1.5$ and angle of attack $\alpha = 0°$, which result in a detached bow shock ahead of the leading edge and two inclined fish-tail shocks emanating from the sharp trailing edge (Figure 5).

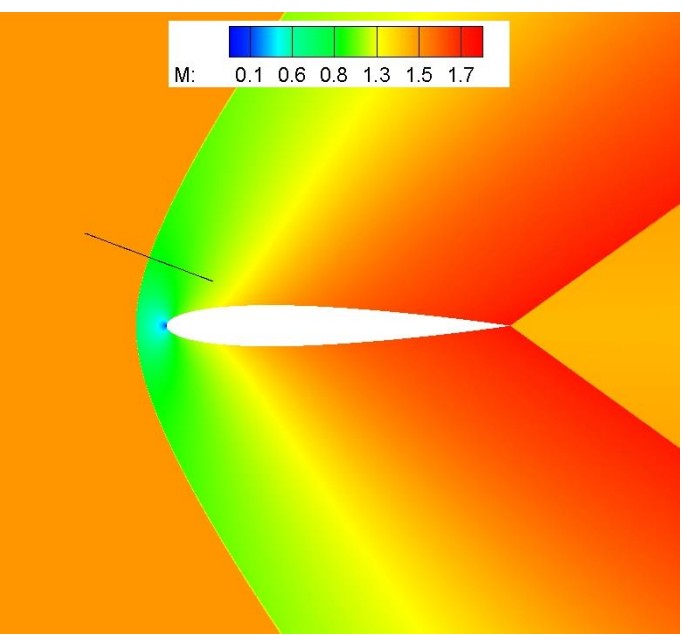

**Figure 5.** Mach contours and cutting line for flow past a NACA0012 airfoil at $M = 1.5$ and $\alpha = 0°$.

Plots of the adjoint variables are shown in Figure 6. Here, and in what follows, the adjoint variables are non-dimensional. Dimensions can be restored by multiplying the variables by the appropriate powers of $\rho_\infty$ and $\left|\vec{v}_\infty\right|$, viz. $\psi_1 = \overline{\psi}_1 / \left(\rho_\infty \left|\vec{v}_\infty\right|\right)$, $\psi_{2,3} = \overline{\psi}_{2,3} / \left(\rho_\infty \left|\vec{v}_\infty\right|^2\right)$, and $\psi_4 = \overline{\psi}_4 / \left(\rho_\infty \left|\vec{v}_\infty\right|^3\right)$ (bars denote non-dimensional variables). Notice that the adjoint solution is constant (actually zero) downstream of the two supersonic characteristics that emanate from the trailing edge. In consequence, the adjoint solution is constant (and its gradient, continuous) across the fish-tail shocks.

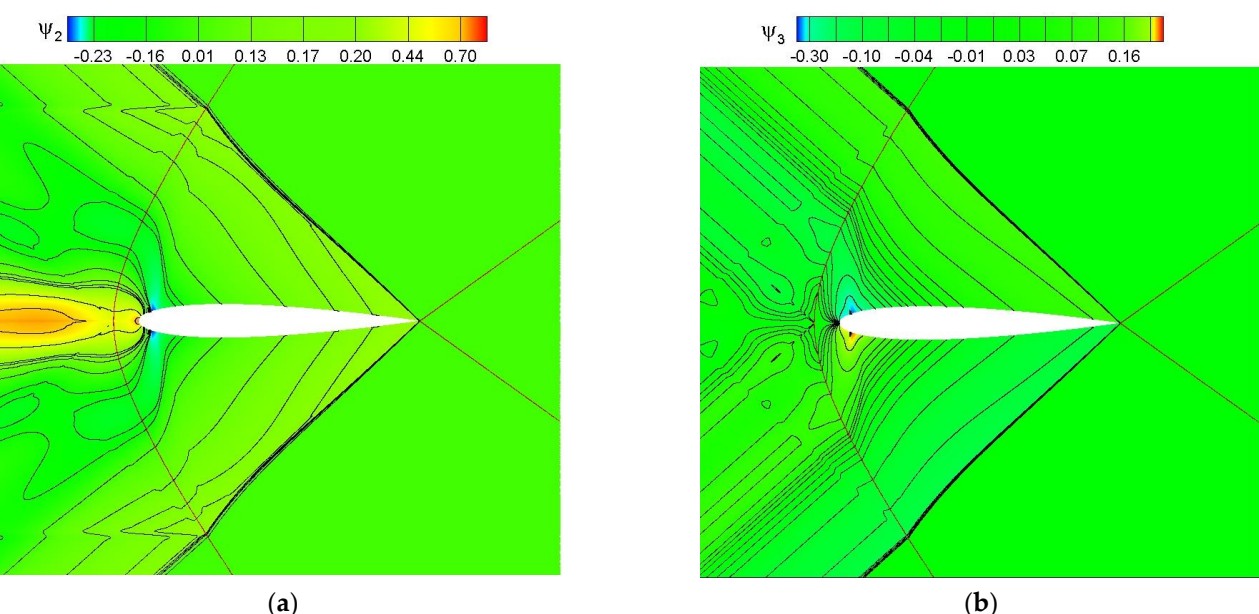

**Figure 6.** Flow past a NACA0012 airfoil at $M = 1.5$ and $\alpha = 0°$. Contour lines for the adjoint x−momentum variable $\psi_2$ (**a**) and y−momentum variable $\psi_3$ (**b**). The bow and fish-tail shocks are indicated for reference.

The solution is continuous across the bow shock as well, but a zoom of the adjoint solution near the shock (Figure 7) shows that the adjoint derivatives are actually discontinuous (notice the abrupt change of direction of the adjoint contour lines of the y-momentum adjoint variable $\psi_3$).

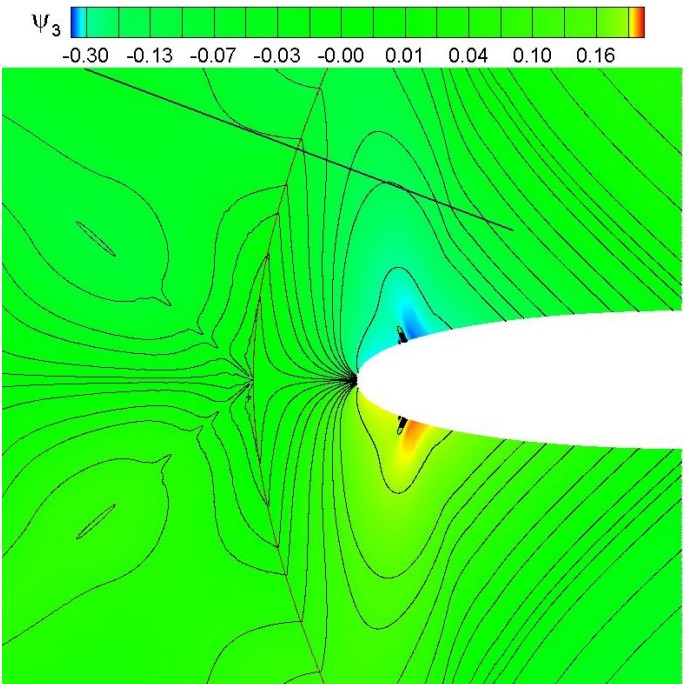

**Figure 7.** Flow past a NACA0012 airfoil at $M = 1.5$ and $\alpha = 0°$. Contour plot for the adjoint y−momentum variable $\psi_3$ near the bow shock. The bow shock and the cutting line are indicated for reference.

We now examine the behavior of the adjoint derivatives along a line crossing the bow shock as indicated in Figures 5 and 7. The line is perpendicular to the shock, which is oblique at this location. The tip of the bow shock is a normal shock, which could be used to test the adjoint properties across such shocks, but the adjoint solution is contaminated by the presence of the adjoint singularity along the incoming streamline stagnation (the supersonic adjoint solution shows singularities along all three characteristics impinging the tip of the bow shock).

Figure 8 shows that the adjoint variables are clearly continuous across the shock, but their gradients are not. A clear kink in the y-momentum adjoint variable $\psi_3$ is visible at the location of the shock. The discontinuity in the normal adjoint gradients is most clearly seen in Figure 9, which plots the adjoint normal derivatives along the line (left panel). The largest jump is found for $\psi_3$, and the relative sizes of the jumps are in agreement with Equation (29). Figure 9 also checks on the right panel some of the properties of the jumps of the adjoint gradients presented in Equations (26) and (27), as well as the adjoint shock Equation (17). Bear in mind that what is being plotted is actually the following:

$$\vec{n}_\Sigma \cdot \partial_n \vec{\phi} = \begin{matrix} v_n \partial_n \psi_1 \\ v_n \partial_n \psi_4 \\ n_\Sigma^x \partial_n \psi_2 + n_\Sigma^y \partial_n \psi_3 \\ \rho v_t (\partial_{tg} \psi_1 + H \partial_{tg} \psi_4) + (p + \rho v_t{}^2) \vec{t}_\Sigma \cdot \partial_{tg} \vec{\phi} \end{matrix} . \tag{36}$$

The plotted functions (36) are continuous across the shock, which indicates that the corresponding jumps are zero as required by Equations (17), (26) and (27).



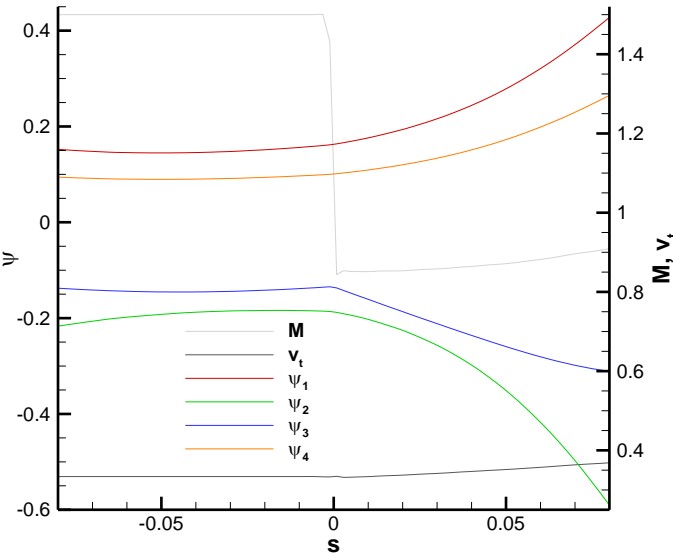

**Figure 8.** Flow past a NACA0012 airfoil at $M = 1.5$ and $\alpha = 0°$. Adjoint variables along the cutting line indicated in Figure 7. Mach number and tangential velocity are also shown for reference.

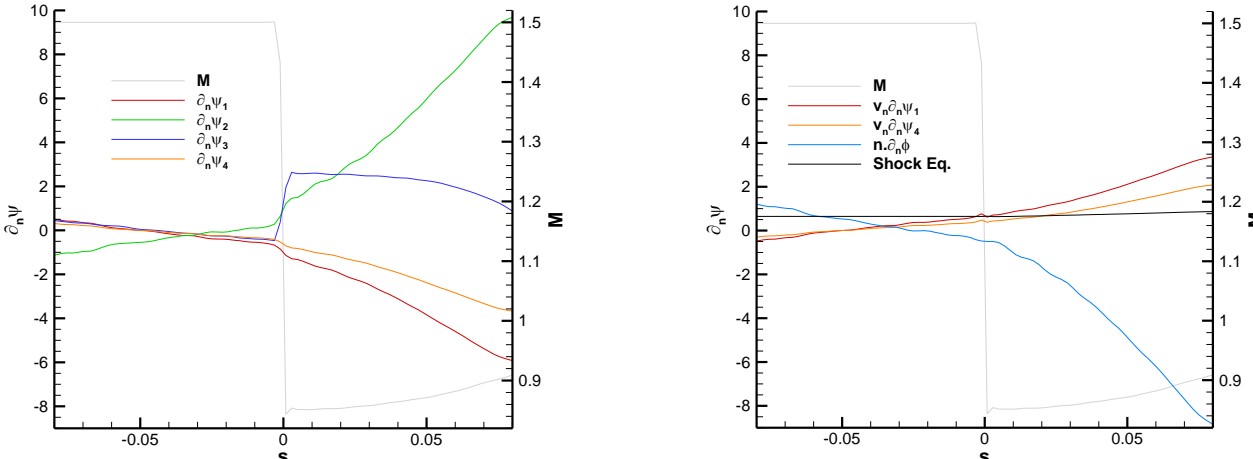

**Figure 9.** Flow past a NACA0012 airfoil at $M = 1.5$ and $\alpha = 0°$. Adjoint normal derivatives and shock relations across the bow shock.

### 4.2. Transonic Flow

The second test case corresponds to transonic flow past a NACA0012 airfoil with flow conditions $M = 0.8$, $\alpha = 1.25°$. Under these conditions, the flow has a fairly strong shock on the upper side at $x/c \approx 0.64$ and a weaker shock on the lower side (Figure 10).

Contour plots of the adjoint variables are shown in Figures 11 and 12. Notice that the adjoint solution is singular along the incoming stagnation streamline [5,21], the wall [21], and also along a delta-like structure formed by the supersonic characteristic emanating from the shock foot and its reflection off the sonic line [22], but it is continuous across the shock and the sonic line. A zoom of the adjoint solution near the upper shock (Figure 12) shows that the adjoint derivatives are actually discontinuous (notice the abrupt change of direction of the adjoint contour lines of the y-momentum adjoint variable $\psi_3$).

We now examine the behavior of the adjoint derivatives along a line crossing the upper shock at right angles as indicated in Figures 10 and 12. As explained above, this shock is not normal (the tangent velocity, though small, is definitely not zero, as can be seen in Figure 13). Figure 13 shows that the adjoint variables are clearly continuous across the shock, but their gradients are not (see the kink in the y-momentum adjoint variable $\psi_3$ at the location of the shock). The discontinuity in the normal adjoint gradients is most

clearly seen in Figure 14, which plots the adjoint normal derivatives along the line (left panel). The largest jump is found for $\psi_3$, while the jumps in the other normal derivatives cannot be appreciated at this level of resolution (and could very well be nearly zero; for $\psi_2$ this is reasonable since, from Equation (29), $[\partial_n \psi_2]_\Sigma \propto t_\Sigma^x = 0$, while for $\psi_1$ and $\psi_4$, it could be related to the fact that $H\partial_n \psi_4 \sim \partial_n \psi_1 \sim v_t \partial_n \psi_3 / 2$ and $v_t$ is relatively small). The right panel of Figure 14 checks the properties of the jumps of the adjoint gradients given in Equations (17), (26) and (27). The plotted functions are continuous across the shock, which indicates that the corresponding jumps are zero.

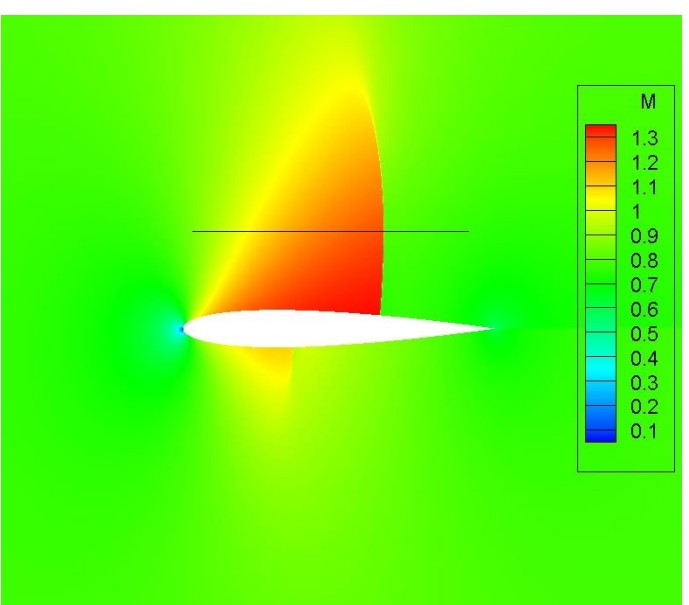

**Figure 10.** Mach contours and cutting line for flow past a NACA0012 airfoil at $M = 0.8$ and $\alpha = 1.25°$.

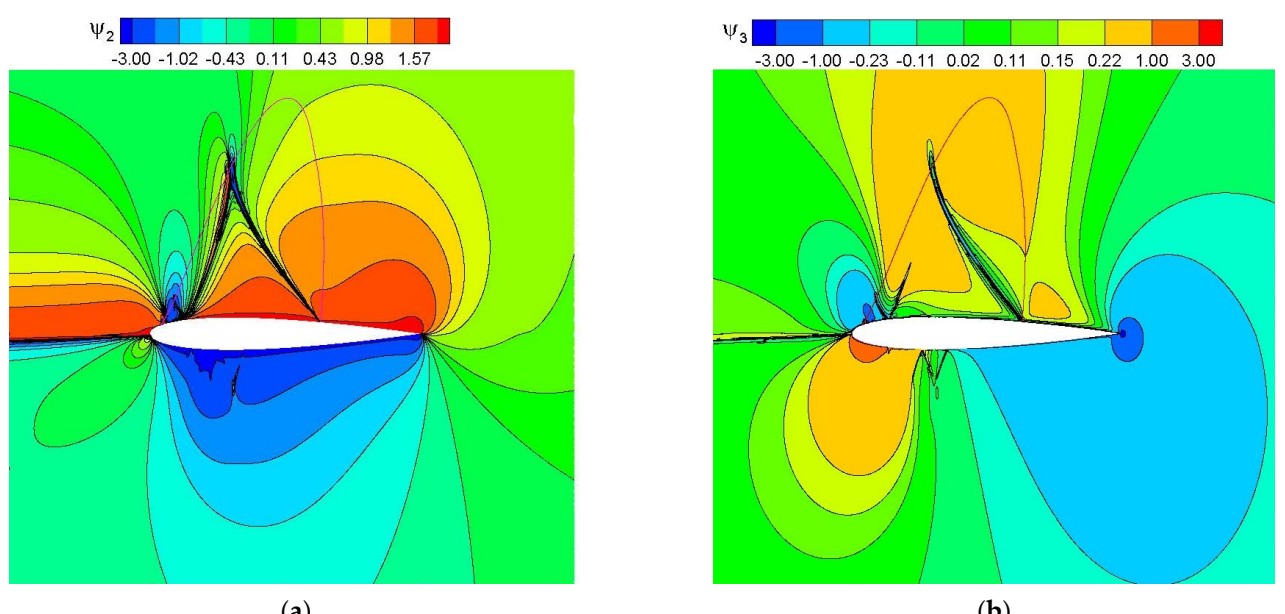

(**a**)                                             (**b**)

**Figure 11.** Flow past a NACA0012 airfoil at $M = 0.8$ and $\alpha = 1.25°$. Contour lines for the x$-$momentum variable $\psi_2$ (**a**) and y$-$momentum variable $\psi_3$ (**b**). The shock position is indicated for reference.

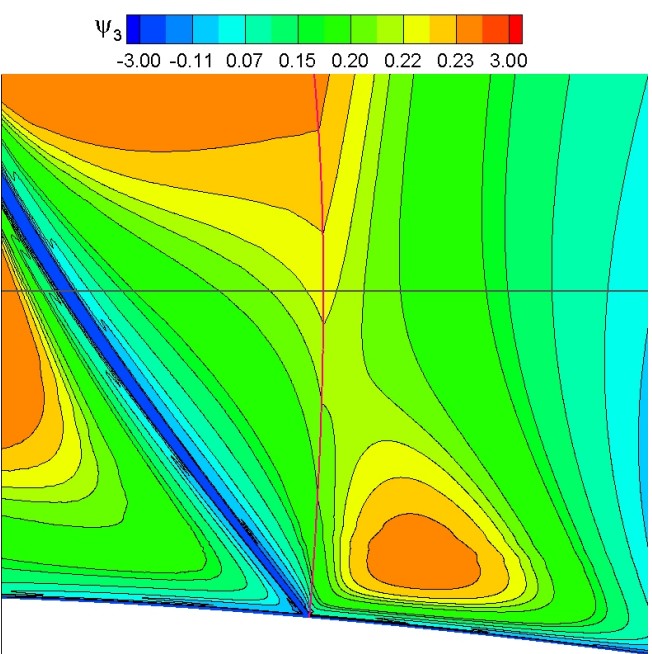

**Figure 12.** Flow past a NACA0012 airfoil at *M* = 0.8 and *α* = 1.25°. Contour plot for the adjoint y−momentum variable $\psi_3$ near the shock. The shock and the cutting line are indicated for reference.

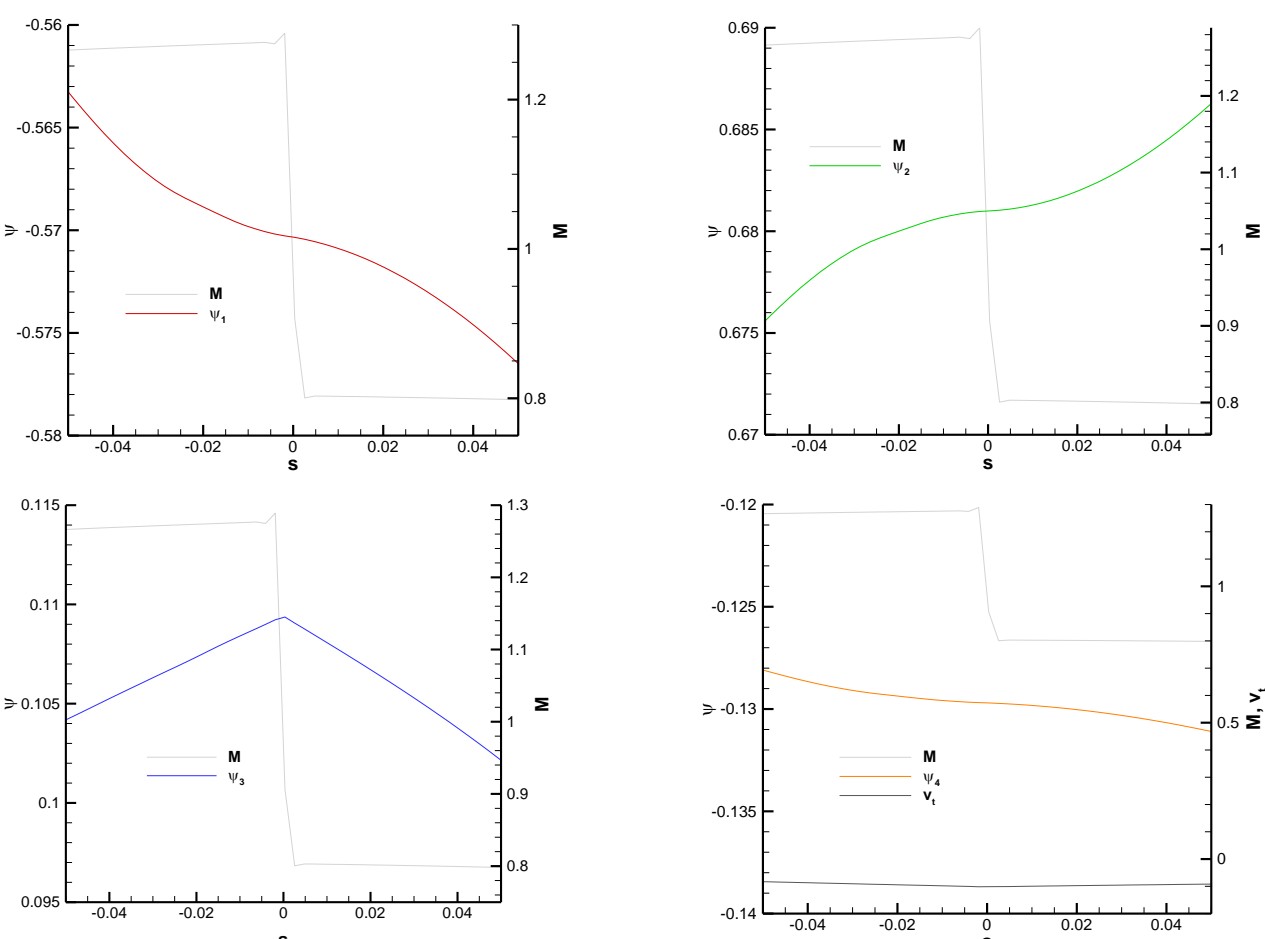

**Figure 13.** Adjoint variables along the cutting line indicated in Figure 12. Mach number and tangential velocity are also shown for reference.

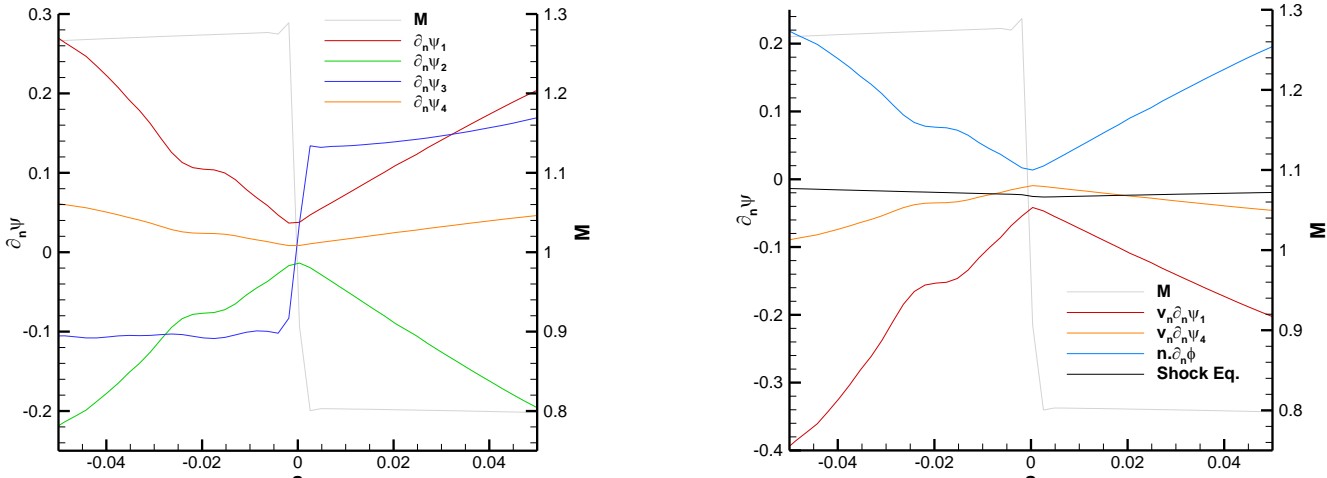

**Figure 14.** Flow past a NACA0012 airfoil at $M = 0.8$ and $\alpha = 1.25°$. Adjoint normal derivatives and shock relations across the shock.

### 4.3. Normal Shock with Normal Extension

For the angle of attack used in the previous case, the upstream wall Mach number $M_1 < M_1^*$ and, thus, the resulting shock is only normal at its foot. We will now consider a special incidence angle $\alpha_{M_1^*}$ for which $M_1 = M_1^*$, which should result in a normal shock to some distance above the surface. The value of $\alpha_{M_1^*}$ is not known a priori and was obtained in [17] through an iterative process by varying the angle of attack while holding $M_\infty$ constant. The final value was, however, not disclosed, so after some numerical experiments, we have found that $\alpha \approx 5.794°$ results in $M_1 \approx M_1^*$ as illustrated in Figure 15.

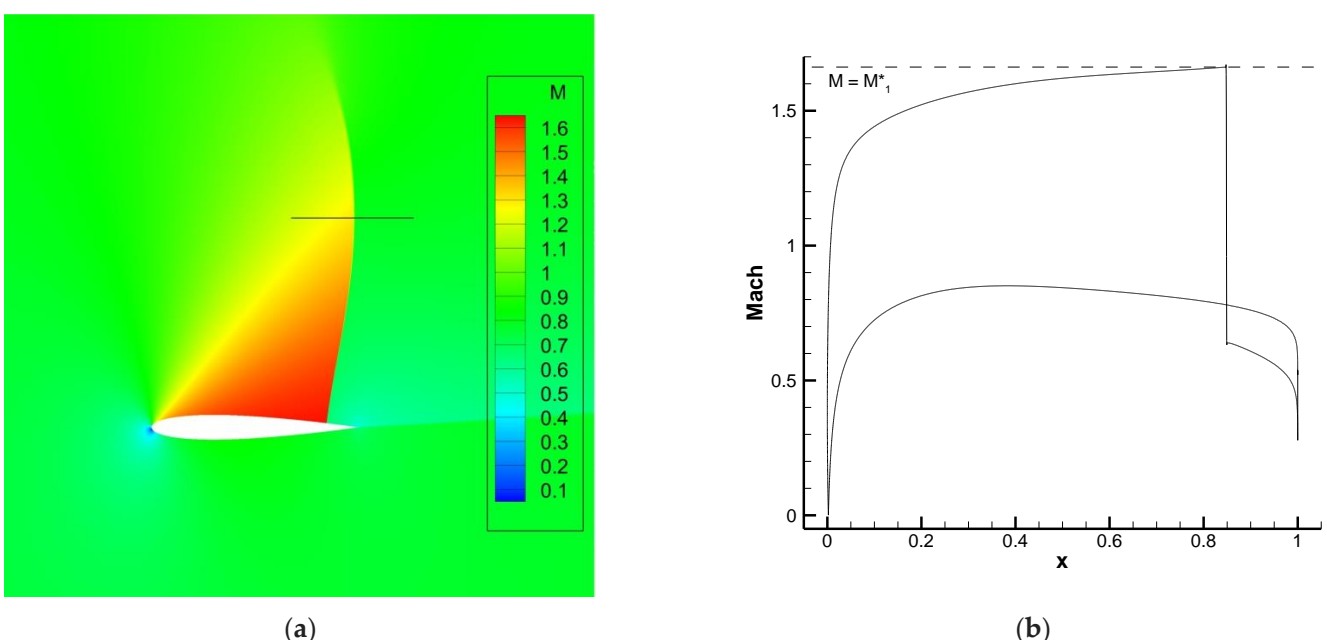

(**a**)　　　　　　　　　　(**b**)

**Figure 15.** (**a**) Mach contours and cutting line and (**b**) surface Mach distribution for flow past a NACA0012 airfoil at $M = 0.8$ and $\alpha = 5.794°$.

At $y = 1$ (in airfoil chord-length units), the shock is still normal, with a fairly low value of $v_t / \left| \vec{v}_\infty \right| \approx 10^{-3}$ (Figure 16). The adjoint variables are still continuous at the shock, but a clear discontinuity in the gradient of $\psi_3$ can be observed in Figure 16 and confirmed by the

plot of the normal derivatives in Figure 17. In the present case, $\vec{t}_\Sigma = (0,1)$, $\vec{n}_\Sigma = (-1,0)$, and $\partial_{tg}\psi^T = (0.29, -0.37, 8 \times 10^{-5}, 0.07)$, which yields, from Equation (34), the prediction:

$$\partial_n\psi^\pm = \begin{pmatrix} 0 \\ 0 \\ \partial_{tg}\psi_2 - (\partial_{tg}\psi_1 + H\,\partial_{tg}\psi_4)/v_n^\pm \\ 0 \end{pmatrix}. \tag{37}$$

We see in Figure 17 that Equation (37) is reasonably obeyed by the numerical solution, as is the other shock relation in Equation (35), $n_\Sigma^x \partial_n\psi_2^\pm + n_\Sigma^y \partial_n\psi_3^\pm = -\partial_n\psi_2^\pm = 0$, and the shock Equation (31), $t_\Sigma^x \partial_{tg}\psi_2 + t_\Sigma^y \partial_{tg}\psi_3 = \partial_{tg}\psi_3 = 0$.

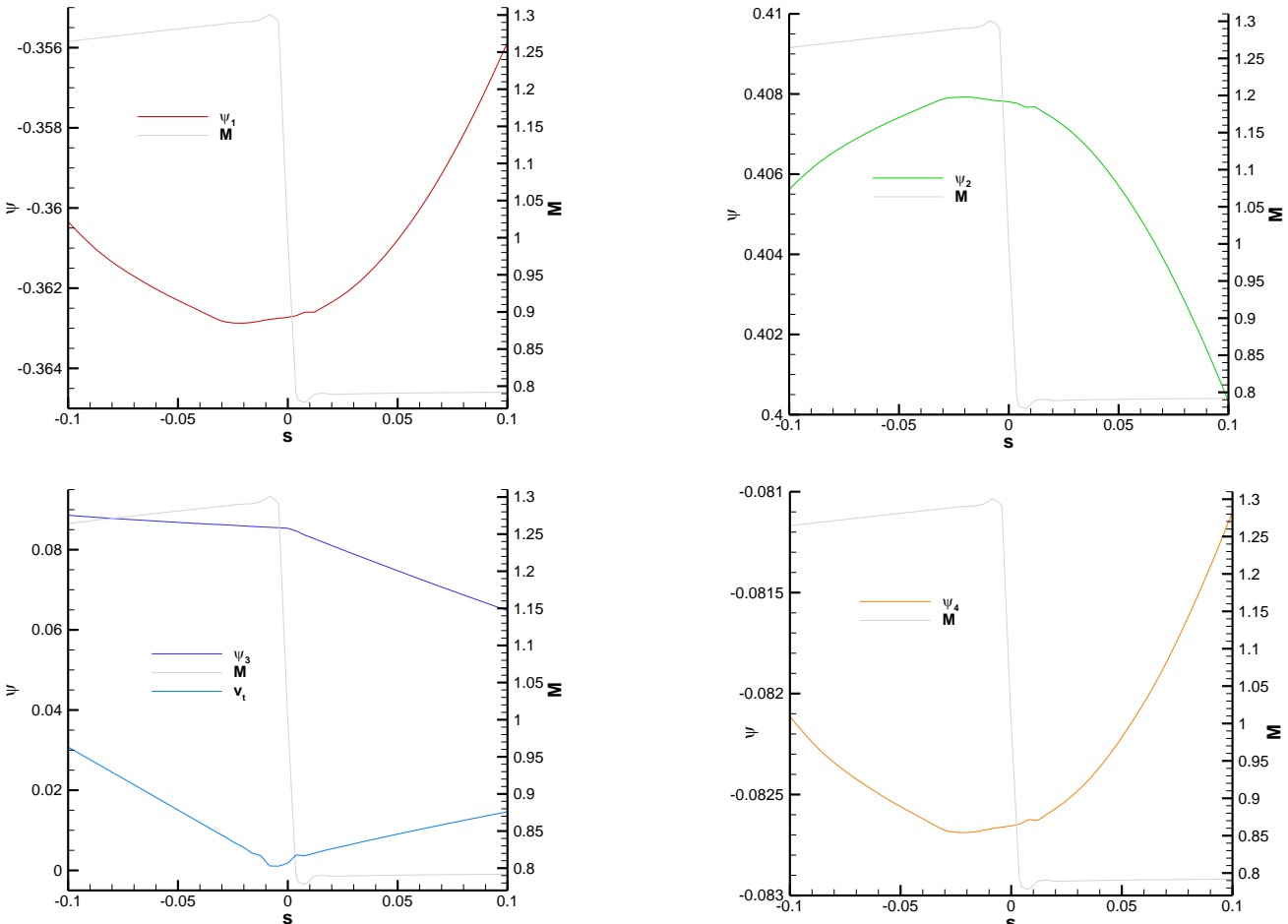

**Figure 16.** Adjoint variables along the cutting line indicated in Figure 15. Mach number and tangential velocity are also shown for reference.

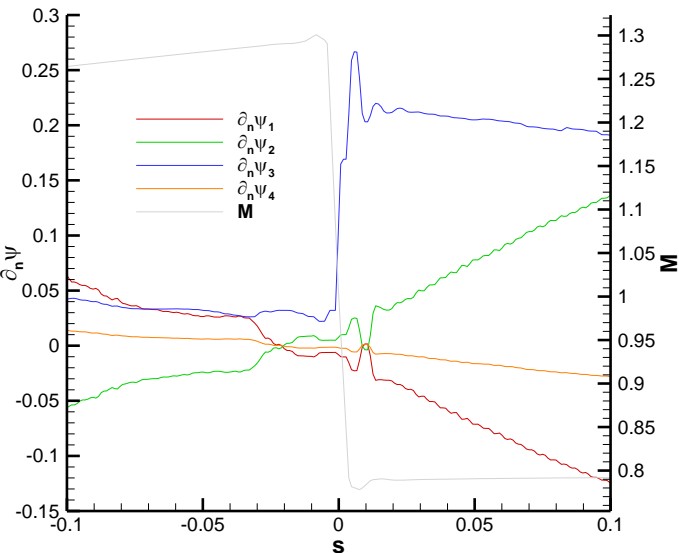

**Figure 17.** Flow past a NACA0012 airfoil at $M = 0.8$ and $\alpha = 5.794°$. Adjoint normal derivatives across the shock.

## 5. Conclusions

In 2D, the adjoint variables of typical cost functions are continuous at shocks and obey a differential equation along the length of the shock. In this work, we have focused on determining the behavior of the gradients of the adjoint variables across shocks. It had been anticipated in [5], by studying the behavior of Green's functions of the linearized Euler equations, that there is a discontinuity in the gradient of the adjoint variables at the location of the shock, which was clearly visible in the numerical results of [15].

In order to analyze the behavior of the adjoint gradients at the shock, it proves convenient to use a frame of reference locally aligned with the shock curve, relative to which it is possible to decompose the adjoint gradients on either side of the shock into (locally) normal and tangent components. Since the adjoint variables are continuous at the shock, the tangent adjoint gradients are continuous as well, but the normal components need not be. Combining the shock equation and the adjoint equations, which hold on either side of the shock, it is possible to obtain detailed predictions for the behavior of the gradients of the adjoint variables across the shock. These equations relate the discontinuity (jump) of the normal component of the adjoint gradients to their tangent component, but do not completely fix the value of the jumps. In fact, the equations allow continuous normal gradients provided that the tangent gradients vanish, as occurs, for example, across the fish-tail shocks in supersonic flows. In 2D, the equations imply that, for normal shocks, three linear combinations of normal adjoint gradients are identically zero at the shock, while a fourth combination has a jump proportional to the discontinuity in the inverse normal component of the velocity $[v_n^{-1}]_\Sigma$.

Several numerical tests have been carried out to get a flavor of how numerical adjoint solutions behave at shocks. The computed solutions are continuous at the shock and present discontinuities in the gradient of the adjoint variables at the location of the shock. The solutions follow the shock relations to a reasonable degree, in agreement with the results presented in [15].

A separate question that we have not addressed here is that of the consistency of the adjoint solution for shocked flows in the limit of infinite grid resolution. In numerical computations, the internal boundary condition (the differential equation along the shock) is not explicitly enforced, which means that there could be errors in the numerical adjoint solution like the ones reported in [10] in 1D cases. The numerical results presented in this work and in [15] do not seem to support this concern, but a detailed grid convergence study would need to be performed to verify this assertion. Results in 1D indicate that

for numerical adjoint results to converge to the analytical solution as the mesh is refined, there must be consistency between the flow and adjoint calculations regarding the level of numerical smoothing (i.e., that one actually uses the same scheme with the same dissipation levels in both cases) and, more critically, that the level of dissipation increases with mesh refinement in such a way that the number of points across the shock increases while at the same time the overall width of the shock decreases. This is relatively simple to achieve in 1D, but the extension to 2D needs to be addressed with care. We hope to come back to these issues in the future.

**Author Contributions:** Both authors have contributed equally to the paper. All authors have read and agreed to the published version of the manuscript.

**Funding:** The research described in this paper has been supported by INTA and the Ministry of Defence of Spain under the grants Termofluidodinámica (IGB99001) and IDATEC (IGB21001).

**Data Availability Statement:** The data presented in this study are available on request from the corresponding author.

**Acknowledgments:** The numerical computations reported in the paper have been carried out with the SU2 code, an open-source platform developed and maintained by the SU2 Foundation.

**Conflicts of Interest:** The authors declare no conflict of interest.

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
