# Peer review of "Shock Equations and Jump Conditions for the 2D Adjoint Euler Equations"

_aerospace, doi:10.3390/aerospace10030267_

Round 1

Reviewer 1 Report

This paper is very interesting and of good quality but it needs some minors correction to be publish.

It is not a good way to cite references to make sentences like this :” For relevant previous work see [1, 2, 3, 4, 5, 6, 7, 8, 9, 10]. “ and it is even worse at the very beginning of an introduction. It must be corrected.

Page 2 there is a footnote and I am note sure it the classical way to do in science article. Moreover references 19 is cited but if it is cited here it should reference 11 I think. Just after in page 3 references 14 is cited and for me it should reference 12. I strongly suggested to the authors to carefully check their references ordering.

 Page 6 same remarks as for page 2 for the footnote

For the results in order to properly validate the method and show its benefits compare to other some comparisons with other simulations and experimental results are dearly needed.

Las but not least the conclusion must be reworked. I must longer and more detailed.

Author Response

We would like to thank you for your review. Below you can find a point-by-point response to your comments. Changes to the manuscript following your suggestions have been highlighted in blue.

Q.- This paper is very interesting and of good quality but it needs some minors correction to be publish.

It is not a good way to cite references to make sentences like this :” For relevant previous work see [1, 2, 3, 4,5, 6, 7, 8, 9, 10]. “ and it is even worse at the very beginning of an introduction. It must be corrected.

R.- This has been corrected.

Q.- Page 2 there is a footnote and I am note sure it the classical way to do in science article.

R.- Footnotes have been removed and incorporated into the main text

Q.- Moreover references 19 is cited but if it is cited here it should reference 11 I think. Just after in page 3 references 14 is cited and for me it should reference 12. I strongly suggested to the authors to carefully check their references ordering.

R.- Thank you. This was a problem with the reference manager, which puts references in footnotes at the end. Since the footnote has been incorporated into the regular text, this problem has been solved and reference ordering is now –hopefully- correct.

Q.- Page 6 same remarks as for page 2 for the footnote

R.- Footnotes have been removed and incorporated into the main text

Q.- For the results in order to properly validate the method and show its benefits compare to other some comparisons with other simulations and experimental results are dearly needed.

R.- We are not sure about what validation results you are suggesting. We are not presenting a method that can be compared to others but a series of equations, or consistency conditions, that constrain the behavior of adjoint variables at shocks. These relations are later on illustrated with numerical solutions. It could be possible to conduct a broader analysis comparing different numerical schemes with regards to the fulfillment of the adjoint shock equations, but that would also require, as explained in more detail in the conclusions, a systematic mesh refinement study with increased dissipation so as to obtain increasedly resolved shocks, all of which is beyond the intended scope of the present work.

Q.- Last but not least the conclusion must be reworked. I must longer and more detailed.

R.- The conclusions have been rewritten.

Reviewer 2 Report

In general, I consider that it is a well written paper research. Nevertheless, I consider that several issues need additional information, and some justifications must be added to improve the quality of the manuscript. It does not imply an outstanding innovation in the approach nor in the methodology in this research field. The differences or contributions that this study makes compared to the previously existing ones should be highlighted, and nothing is found in your obtained results. In this sense you should try to highlight what is the innovation of you approach, and focus the analysis on novelties. You should try to focus the objective on something more specific, it must contribute to the state of the art in this matter. This research is a theoretical approach, focused more on analysis than on application. It seems to me a teaching contribution propitious in the university environment, but I am not sure if it is suitable for this journal, in more detail:.

The extension of the abstract is shorter than usual. In my opinion it should be extended.

Introduction. The objective of this research was not clearly defined. It is crucial from my point of view to understand the work itself. The objectives, approach, and methodology are not clearly defined, the research is not well focused.

Line 15. 'For relevant previous work see [1-10]’. This is a not specific reference. It must be avoided. Too many grouped references have been included. It is not correct, you must be more specific.

All the equations must be numbered, and relative references included in the text explanations. Some of the equations are numbered but not all of them, it is confusing, please modify accordingly.

Line 80. Some Chinese symbols are overwritten over the inline equations, please review.

Lines 247-248. ‘a very fine unstructured mesh with 5.2×106 nodes and 10.4×106 triangles, with 6400 nodes on the airfoil profile.’ Is the size of the used mesh enough? there is no mesh convergence study, it should be done. It is very important.

Figures 5 and 6. The magnitude of values represented, scale, and colour map are not explained. It must be improved. The units of all parameters must be included, maybe you can overdraw contours and other things in a more understandable way.

Line 283. ‘gradients is most clearly seen in Figure 8’. Please check English carefully.

Lines 349-351. ’The final value was however not disclosed, so after some numerical experiments increasing the incidence across different mesh levels we have found that …’ A mesh convergence study must be done, and before. It is not adequate. The mesh parameters are not sufficiently defined, and in general I cannot be sure about the mesh quality. You cannot ensure that the results shown correspond to an adequate convergence. I think there is a broad general consensus on this aspect in the numerical simulation community. The values of cell sizes in the near wall region, first layer thickness, the boundary layer mesh details, the number of layers, etc. are not specified, and this information is a crucial information, and there is also no image showing the mesh. Is there any refinement in the wake area?, it is important too. It is not enough to specify the global cells number. The mesh used is not sufficiently well justified, and in general I have serious doubts about the mesh convergence study.

Can you give some information about the used computers? How many nodes are being used? What was the total time for which the simulation was run? You don’t specify, all this information must be indicated.

Author Response

We would like to thank you for your review. Below you can find a point-by-point response to your comments. Changes to the manuscript following your suggestions have been highlighted in green.

Q.- In general, I consider that it is a well written paper research. Nevertheless, I consider that several issues need additional information, and some justifications must be added to improve the quality of the manuscript. It does not imply an outstanding innovation in the approach nor in the methodology in this research field. The differences or contributions that this study makes compared to the previously existing ones should be highlighted, and nothing is found in your obtained results. In this sense you should try to highlight what is the innovation of you approach, and focus the analysis on novelties. You should try to focus the objective on something more specific, it must contribute to the state of the art in this matter. This research is a theoretical approach, focused more on analysis than on application. It seems to me a teaching contribution propitious in the university environment, but I am not sure if it is suitable for this journal, in more detail:.

The extension of the abstract is shorter than usual. In my opinion it should be extended.

R.- The abstract has been extended

Q.- Introduction. The objective of this research was not clearly defined. It is crucial from my point of view to understand the work itself. The objectives, approach, and methodology are not clearly defined, the research is not well focused.

R.- Both the abstract and introduction state that the objective of the research is the behavior of adjoint solutions at shock waves. This is a research topic that has received considerable attention in the past, as the references to the literature show. In this regard, the introduction carefully tries to position the present paper with respect to previous works.

Q.- Line 15. 'For relevant previous work see [1-10]’. This is a not specific reference. It must be avoided. Too many grouped references have been included. It is not correct, you must be more specific.

R.- The references have been separated

Q.- All the equations must be numbered, and relative references included in the text explanations. Some of the equations are numbered but not all of them, it is confusing, please modify accordingly.

R.- Done

Q.- Line 80. Some Chinese symbols are overwritten over the inline equations, please review.

 R.- We are sorry. This must be a problem with the internal conversion in the journal’s servers. The paper looks fine on our side.

 Q.- Lines 247-248. ‘a very fine unstructured mesh with 5.2×106 nodes and 10.4×106 triangles, with 6400 nodes on the airfoil profile.’ Is the size of the used mesh enough? there is no mesh convergence study, it should be done. It is very important.

R.- Thank you. Further details about the mesh have been added. A comment has been also included referencing the used mesh with respect to the state of the art regarding mesh size for this type of inviscid computations (Vassberg & Jameson and Peter et al.). A comment about mesh convergence studies has been also included in the conclusions, and we elaborate on this in our response to your final comments (see below)

Q.- Figures 5 and 6. The magnitude of values represented, scale, and colour map are not explained. It must be improved. The units of all parameters must be included, maybe you can overdraw contours and other things in a more understandable way.

R.- The plots have been reworked to show the scale of the variables. It is stated in pag. 11 that the adjoint variables shown in the plots are non-dimensional, and instructions are given to restore full dimensions if necessary. This comment was before in a footnote and has been incorporated into the main text.  

 Q.- Line 283. ‘gradients is most clearly seen in Figure 8’. Please check English carefully.

R.- Thank you. In the complete sentence “The discontinuity in the normal adjoint gradients is most clearly…” the subject is “discontinuity”, not gradients, so we believe that the verb must be singular, not plural.  

Q.- Lines 349-351. ’The final value was however not disclosed, so after some numerical experiments increasing the incidence across different mesh levels we have found that …’ A mesh convergence study must be done, and before. It is not adequate. The mesh parameters are not sufficiently defined, and in general I cannot be sure about the mesh quality. You cannot ensure that the results shown correspond to an adequate convergence. I think there is a broad general consensus on this aspect in the numerical simulation community. The values of cell sizes in the near wall region, first layer thickness, the boundary layer mesh details, the number of layers, etc. are not specified, and this information is a crucial information, and there is also no image showing the mesh. Is there any refinement in the wake area?, it is important too. It is not enough to specify the global cells number. The mesh used is not sufficiently well justified, and in general I have serious doubts about the mesh convergence study.

R.- The sentence you are mentioning does not refer to a mesh convergence study but to the approach we have used to obtain an approximately normal shock with normal extension. It is not the object of this paper to study the physics of such shocks but to run a 2D case with a normal shock wave, or the closest to normal that we could find, and on view of the small value of the tangent velocity across the shock we believe we have succeeded in that. The mention about the different mesh levels was just to give a flavor about the process to reach the value of the incoming Mach number for the critical case. The authors of the referenced work (B. Koren and E. van der Maarel, "On steady, inviscid shock waves at continuously curved, convex surfaces," Theoret. Comput. Fluid Dynamics, vol. 4, p. 177–195, 1993) used a Newton algorithm to establish the value of the incoming Mach number). This value depends on both the mesh and the dissipation scheme, and their working mesh was relatively small. Repeating that approach with our 5 M node mesh would be extremely time-consuming, so we preferred to start scanning for the actual value on coarse meshes and reserve the final fine-tuning for the actual mesh. In any case, we have erased that comment as it may confuse the readers and it is not instrumental to the objectives of the paper.

As for the whole issue of the mesh quality and its adequacy, we have already made a few remarks in our response to one of your previous questions. Elaborating on that, please note that this is an inviscid computation and that we are using a uniformly refined, 5 million point mesh with 6400 nodes on the airfoil profile for two-dimensional, inviscid computations. This is in perfect alignment with the published literature. J. Peter et al (Journal of Computational Physics, 449, 110811, 2022), for example, show results on 2049x2049 and 4079x4079 meshes (albeit with a fluid domain larger than ours and with fewer nodes on the airfoil profile), while according to Vassberg & Jameson, “In Pursuit of Grid Convergence for Two-Dimensional Euler Solutions”, J. Aircraft 47(4), 2010, 1152-1166, asymptotic ranges for conventional, finite-volume, second order solvers of the type used in this paper, are reached at around one quarter of a million cells at the latest. It is also to be noted that the meshes used in those works are uniformly refined and do not target special regions such as shocks or wakes.

We have also included the size of the near-wall elements, which is 10-5 in chord-length units, which is about the typical near-wall resolution required for turbulent 2D computations on airfoils. At any rate, a flavor of the near-wall resolution can be extracted from the quoted number of nodes on the airfoil profile, which is 6400. Details about the boundary layer mesh are not provided as they are of no real concern in inviscid computations.

Finally, regarding the mesh convergence study, please note that our numerical results are used to show how typical adjoint solutions obtained with typical, well-known, widely used and profusely validated solvers behave in shocked flows with regards to the analytic shock relations presented in the paper that constitute the heart and object of the research. There is no attempt to impeach or judge the analytical results based on the numerical results. This is the same approach taken, for example, in J. Peter et al (op. cit.).

A mesh convergence study in this context would be interesting and should focus on determining whether and under which circumstances numerical adjoint solutions converge to the analytic solution in the limit of infinite mesh resolution. This has been analysed by Giles and Giles & Ulbrich (op. cit.) for 1D cases, and the conclusion is that the level of dissipation across the shock in both the flow and adjoint solvers has to be consistent and increase with mesh refinement in such a way that the number of points across the shock increases while at the same time the overall width of the shock decreases. This is relatively easy to do in 1D, but how this translates to 2D remains, in our opinion, an open issue and, surely, the object of future work. A comment on this regard has been added to the conclusions.

Q.- Can you give some information about the used computers? How many nodes are being used? What was the total time for which the simulation was run? You don’t specify, all this information must be indicated.

R.- The computations have been carried out on a cluster of Intel Xeon processors. For typical runs we have utilized as many as 192 cores and those took about 30 h to converge the critical Mach number transonic adjoint case to machine precision. We have not provided this information as we understand it to be irrelevant for the research presented in the paper. No new numerical method or algorithm is presented whose efficiency has to be assessed, and the details and performance of the software used (Stanford University’s SU2 solver), which we have not developed ourselves, is well documented in the literature.

Round 2

Reviewer 2 Report

First of all, I would like to acknowledge the effort made by the authors in the detailed and extensive answers, and the effort to include some modifications and clarifications in the document. I consider that the quality of the paper has been improved, and now it could be published.